

# Hepatic nuclear factor 4 alpha promotes the ferroptosis of lung adenocarcinoma *via* transcriptional activation of cytochrome P450 oxidoreductase

Valeria Besskaya\*, Huan Zhang\*, Yunyi Bian, Jiaqi Liang, Guoshu Bi, Guangyao Shan, Cheng Zhan and Zongwu Lin

Department of Thoracic Surgery, Zhongshan Hospital, Fudan University, Shanghai, China
\* These authors contributed equally to this work.

Corresponding authors
Cheng Zhan, czhan10@fudan.edu.cn
Zongwu Lin,
lin.zongwu@zs-hospital.sh.cn

## ABSTRACT

**Background:** Lung adenocarcinoma is one of the most prevalent cancers while ferroptosis is crucial for cancer therapies. This study aims to investigate the function and mechanism of hepatic nuclear factor 4 alpha (HNF4A) in lung adenocarcinomas' ferroptosis.

**Materials and Methods:** HNF4A expression in ferroptotic A549 cells was detected. Then HNF4A was knocked down in A549 cells while overexpressed in H23 cells. Cells with changed HNF4A expression were tested for cytotoxicity and the level of cellular lipid peroxidation. The expression of cytochrome P450 oxidoreductase (POR) expression was examined after HNF4A was knocked down or overexpressed. Chromatin immunoprecipitation-quantitative PCR (ChIP-qPCR) and dual-luciferase assays were performed to validate the regulation of HNF4A on POR. Finally, POR was restored in HNF4A-altered cells to check whether it restores the effect of HNF4A on ferroptosis.

**Results:** We found that HNF4A expression significantly decreased in the ferroptosis of A549 cells, and this change can be blocked by deferoxamine, an inhibitor of ferroptosis. Knockdown of HNF4A inhibited ferroptosis in A549 cells while overexpression of HNF4A promoted ferroptosis in H23 cells. We identified a key ferroptosis-related gene, POR serves as a potential target gene of HNF4A, whose expression was significantly changed in lung adenocarcinoma cells knocking down or overexpressing HNF4A. We demonstrated that HNF4A was bound to the POR's promoter to enhance POR expression, and identified the binding sites *via* ChIP-qPCR and luciferase assays. Restoration of POR expression blocked the promoting effect of HNF4A on ferroptosis in lung adenocarcinoma.

**Conclusion:** HNF4A promotes POR expression through binding to the POR's promoter, and subsequently promotes the ferroptosis of lung adenocarcinoma.

## INTRODUCTION

As one of the most prevalent malignant tumors in recent years, lung cancer has the second- and first-highest morbidity and mortality of all cancers (*Sung et al., 2021*). Lung

adenocarcinoma, which makes up nearly two-thirds of all new occurrences of lung cancer, is currently the most common pathological subtype of lung cancer (*Lu et al., 2019*). Surgery, target treatment, chemotherapy, radiation, and immunotherapy are the cornerstones of lung cancer clinical care. Yet, because of its powerful ability to proliferate and secondary resistance, lung adenocarcinoma still has insufficient treatment efficacy. The frontier problems of current research continue to be identifying novel treatment targets and improving the therapeutic effect of lung cancer.

Ferroptosis is a novel form of iron-dependent programmed cell death (*Chen et al., 2021*; *Stockwell et al., 2017*). The primary mechanism is that external stimuli lead to an imbalance of reactive oxygen species (ROS) in cells, which triggers the iron-catalyzed Fenton reaction. This reaction changes the structure of phospholipid bilayers, causes biofilms to rupture, and ultimately results in cell death (*Tang et al., 2021*).

As ferroptosis plays a significant role in many pathophysiological processes, it can be carefully controlled to prevent the onset of a number of disorders. On the one hand, lowering ferroptosis may prevent the pathological apoptosis of myocardial, glomerulus, nerve, and other cells in conditions like organ ischemia-reperfusion injury and neurological disorders (*Capuzzimati, Hough & Liu, 2022*; *Yan et al., 2021b*). On the other hand, it is possible to use ferroptosis in tumors to prevent tumor development and improve the outcomes of radiation therapy, chemotherapy, and immunotherapy. For instance, *Lei et al. (2020)* suggested a novel approach to increase the efficacy of radiation by combining radiotherapy with agents that produce ferroptosis. According to *Bi et al. (2022)* ferroptosis was crucial in overcoming cisplatin and pemetrexed resistance in lung cancer. Ferroptosis may also be useful in immunotherapy since CD8+ T lymphocytes activated during immunotherapy cause ferroptosis in tumor cells *via* IFN-γ, *etc.* (*Liao et al., 2022*; *Wang et al., 2019*).

Thus, in order to enhance the therapeutic effects of lung adenocarcinoma, it is essential to investigate the molecular processes of ferroptosis. In this study, we discovered that a transcript factor called hepatocyte nuclear factor 4 alpha (HNF4A) dramatically increased cytochrome P450 oxidoreductase (POR), which in turn significantly accelerated ferroptosis in lung adenocarcinoma. Promoting HNF4A-regulated ferroptosis may be beneficial in the clinical treatment of lung adenocarcinoma.

## MATERIALS AND METHODS

### Cell culture

The Chinese National Collection of Authenticated Cell Cultures (Shanghai, China) provided the lung adenocarcinoma cell lines A549 and H23, and human embryonic kidney cell line 293T. Cells were grown in 10% fetal bovine serum-containing DMEM medium (with high glucose) (KeyGEN, Nanjing, Jiangsu, China).

### Lentiviruses and compounds

GENECHEM (Shanghai, China) provided the negative control virus, sh-HNF4A-1 (5′- GCTTCTCTCCAAGGCTAAACT -3′), sh-HNF4A-2 (5′- GGACTAGCAAACTCTACAAAT-3′), and HNF4A overexpression viruses. (1S,3R)-

RSL3 was purchased from MedChemExpress (NJ, USA), while imidazole ketone erastin (IKE) and deferoxamine (DFO) were both purchased from TOPSCIENCE (Shanghai, China).

## Cytotoxicity assay

To measure the cell sensitivity to ferroptosis inducers, a total of 4,000 cells were seeded into each well of 96-well plates and subsequently treated with 0, 1, 2, 5, and 10 uM RSL3, or 0, 2, 5, 10, and 20 uM IKE. Then 10 ul alamaBlue (YEASEN, Shanghai, China) was added 48 h after treatments in each well. After 2 h of incubation, the fluorescence was monitored at 545 nm excitation and 590 nm emission wavelengths to reflect the cells' sensitivity as previously described (*Liang et al., 2023*).

## Lipid peroxidation assay

Cells were pretreated with 8 uM RSL3 or 16 uM IKE for 24 h, and then digested to single cell suspensions, incubated with 4 mM BODIPY 581/591 C11 (Thermo Fisher, Waltham, MA, USA) for 30 min at 37 °C. Cells' fluorescence intensity was detected within the FITC channel using FACSAria III flow cytometry (BD, Franklin Lakes, NJ, USA), and then analyzed using FlowJo software (version VX, OR, USA).

## Quantitative real-time PCR (qRT-PCR)

TRNzol Universal (TIANGEN, Beijing, China), Hifair® III 1st Strand cDNA Synthesis SuperMix for qPCR (YEASEN, Shanghai, China), and Hieff® qPCR SYBR Green Master Mix (Low Rox Plus) (YEASEN, Shanghai, China) were used to perform qRT-PCR as previously described (*Liang et al., 2022*). The expression was calculated using the $2^{-\Delta\Delta CT}$ method, with β-actin serving as the reference gene. These were the primers were as follows: HNF4A-F:5′- CGAAGGTCAAGCTATGAGGACA -3′, HNF4A-R:5′- ATCTGCGATGCTGGCAATCT -3′ POR-F: 5′- CGGAACCACGCACTTTCATTTCTC -3′, and POR-R: 5′- TGCCAGCGGTGAGTGCTATCT-3′, β-actin-F: 5′- TGACGTGGACATCCGCAAAG-3′, β-actin-R: 5′- CTGGAAGGTGGACAGCGAGG-3′.

## Western blot assay

Cells were lysed using RIPA Buffer (Beyotime, Jiangsu, China) containing 1 mM PMSF (Beyotime, Jiangsu, China). BCA Protein Quantification Kit (YEAEN) was used to determine the protein concentration. A total of 20 ug protein was separated using SDS-PAGE and then transferred to PVDF films. We utilized the HNF4A (1:1,000, CY10632; Abways, Shanghai, China) and POR (1:1,000, ab180597; Cambridge, UK) antibodies to investigate the protein expression, and GAPDH (1:3,000, AG019; Beyotime, Jiangsu, China) as the reference.

## Chromatin immunoprecipitation (ChIP) assay

To examine the binding area in the POR's promoter, we used ChIP-Seq data of HNF4A from the Encode database (https://www.encodeproject.org), and analyzed the data using the Integrative Genomics Viewer software (version 2.13.1, https://software.broadinstitute.org/software/igv/). Then, we employed ChIP-qPCR to validate the binding of HNF4A in

the promoter of POR using SimpleChIP® Plus Enzymatic Chromatin IP Kit (Magnetic Beads) (CST, Boston, MA, USA) and HNF4A Antibody (1:50, CY10632; Abways, New York, NY, USA). The primers used in ChIP-qPCR have the following sequences: primer_F: 5′- CGGAACCACGCACTTTCATTTCTC -3′, and primer_R: 5′- TGCCAGCGGTGAGTGCTATCT-3′.

## Dual-luciferase reporter assay

The −1,000 to 200 bp of the POR's promoter sequence (NM_001367562.3) was cloned into the firefly luciferase reporter plasmid after being retrieved from the National Center for Biotechnology Information database (https://www.ncbi.nlm.nih.gov/gene/). The probable HNF4A binding sequences in the POR's promoter region were predicted using the JASPAR website (https://jaspar.genereg.net). Subsequently, another firefly luciferase reporter plasmid was cloned with the mutant POR's promoter sequence, which had both probable HNF4A binding sites mutated. We transfected these plasmids into 293T cells with or without HNF4A overexpression to perform the dual-luciferase reporter assay using the Lipo8000 Transfection Reagent (Beyotime, Jiangsu, China) and the Luciferase Reporter Gene Assay Kit (Beyotime, Jiangsu, China) according to the manufacturer's instruction. Renilla luciferase reporter plasmid was employed as the reference. The creation of each plasmid was handled by CENECHEM.

## Statistical analyses

Student's $t$-test and ANOVA tests were run using R software (version 4.0.3) to compare variables. A two-tail of $p < 0.05$ was regarded as significant. Bonferroni adjustment is used where multiple hypothesis testing is performed.

# RESULTS

## HNF4A expression significantly decreased in ferroptotic A549 cells

We discovered that the expression of HNF4A decreased by more than 50% in ferroptosis based on previously reported RNA-Seq data of A549 cells treated with the ferroptosis inducers RSL3 or IKE (Bi et al., 2022). We treated A549 cells with RSL3 and IKE with or without the ferroptosis inhibitor DFO, and then utilized qRT-PCR and western blot to detect the changes in HNF4A expression, in order to further confirm the decrease in HNF4A expression brought on by ferroptosis. The reduction in HNF4A expression brought on by ferroptosis inducers was prevented by DFO administration, as shown in Figs. 1B and 1C, indicating that HNF4A may be directly engaged in the cellular molecular changes in ferroptosis.

## HNF4A promoters ferroptosis and POR's expression in lung adenocarcinoma

To explore the function of HNF4A in ferroptosis, we sought to modify HNF4A expression in lung adenocarcinoma cells. Gene expression profiles of multiple commonly used lung adenocarcinoma cell lines were downloaded from the Cancer Cell Line Encyclopedia (CCLE) database (https://sites.broadinstitute.org/ccle/). As shown in Fig. 2A, HNF4A was

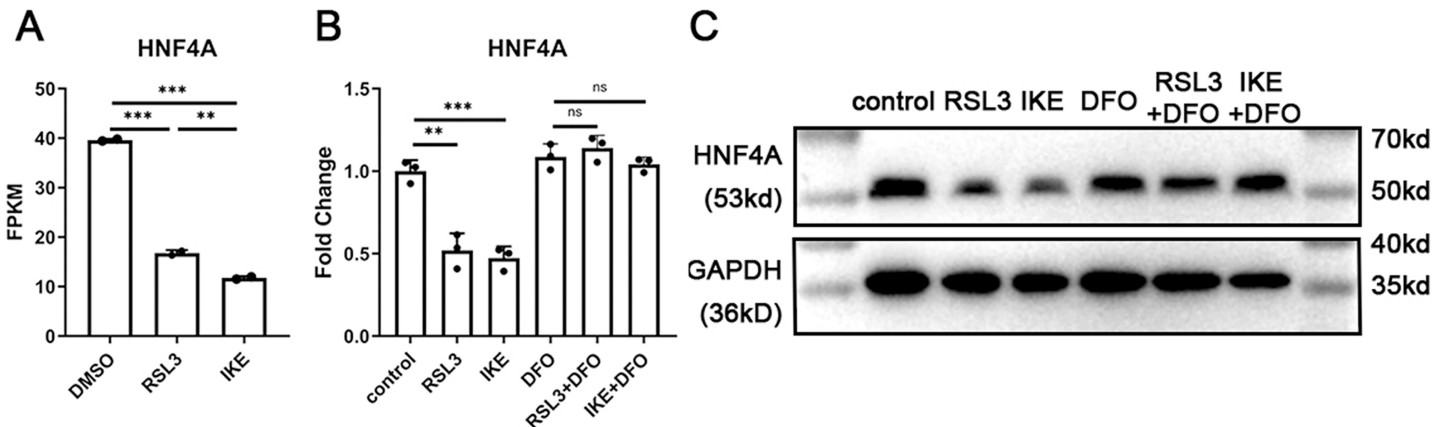

**Figure 1 HNF4A expression significantly decreased in ferroptotic A549 cells.** (A) The HNF4A expression in A549 cells treated with RSL3 and IKE. (B) qRT-PCR and Western blot (C) results of HNF4A expression in A549 cells after the treatment of RSL3 and IKE, with or without DFO. ns, not significant; *$p < 0.05$, **$p < 0.01$, ***$p < 0.001$.                               

strongly expressed in A549 and CALU3 cell lines and rarely expressed in H23, H358, H1650, H1975, HCC827, and PC9 cell lines.

Therefore, we knocked down the expression of HNF4A in A549 cells and overexpressed it in H23 cells (Figs. 2B and 2C), and investigated whether or not the cells' susceptibility to inducers of ferroptosis changed. As shown in Fig. 2D, HNF4A knockdown significantly decreased the cytotoxicity of RSL3 and IKE to A549 cells, whereas HNF4A overexpression increased the lethality of ferroptosis inducers on H23 cells. In parallel, HNF4A knockdown increased the level of cellular lipid peroxidation in lung adenocarcinoma cells treated with ferroptosis inducers, while HNF4A overexpression had the opposite effect (Fig. 2E).

HNF4A, as a transcript factor, may promote ferroptosis by regulating the expression of its downstream target genes. Therefore, we obtained the data about HNF4A's potential target gene from the gene transcription regulation database (GTRD) (http://gtrd.biouml. org/), and looked for ferroptosis-related genes among the potential target genes of HNF4A. We discovered that HNF4A most likely regulated the expression of POR, one of the key ferroptosis-promoting genes, with the highest peak count data in the GTRD data. Also, our qRT-PCR and western blot results demonstrated that HNF4A knockdown significantly reduced POR expression whereas HNF4A overexpression promoted POR expression in lung adenocarcinoma cells (Figs. 2B and 2C).

## HNF4A promoted POR expression *via* binding to POR's promoter

We analyzed the ChIP-seq data for HNF4A in the Encode database, and found that there were several binding peaks of HNF4A in the promoter region of POR (Fig. 3A). Using JARSPAR, we identified two potential binding sites for HNF4A within 100 bp before the POR transcription start site (TSS) (Fig. 3B), and designed a pair of primers around the two binding sites. Our ChIP-qPCR data confirmed that HNF4A was bound to the POR's promoter region in A549 and H23 cells, as shown in Fig. 3C.

Dual luciferase assays were subsequently used to validate the regulation of POR's expression by HNF4A. We created two different firefly luciferase plasmids: one with the

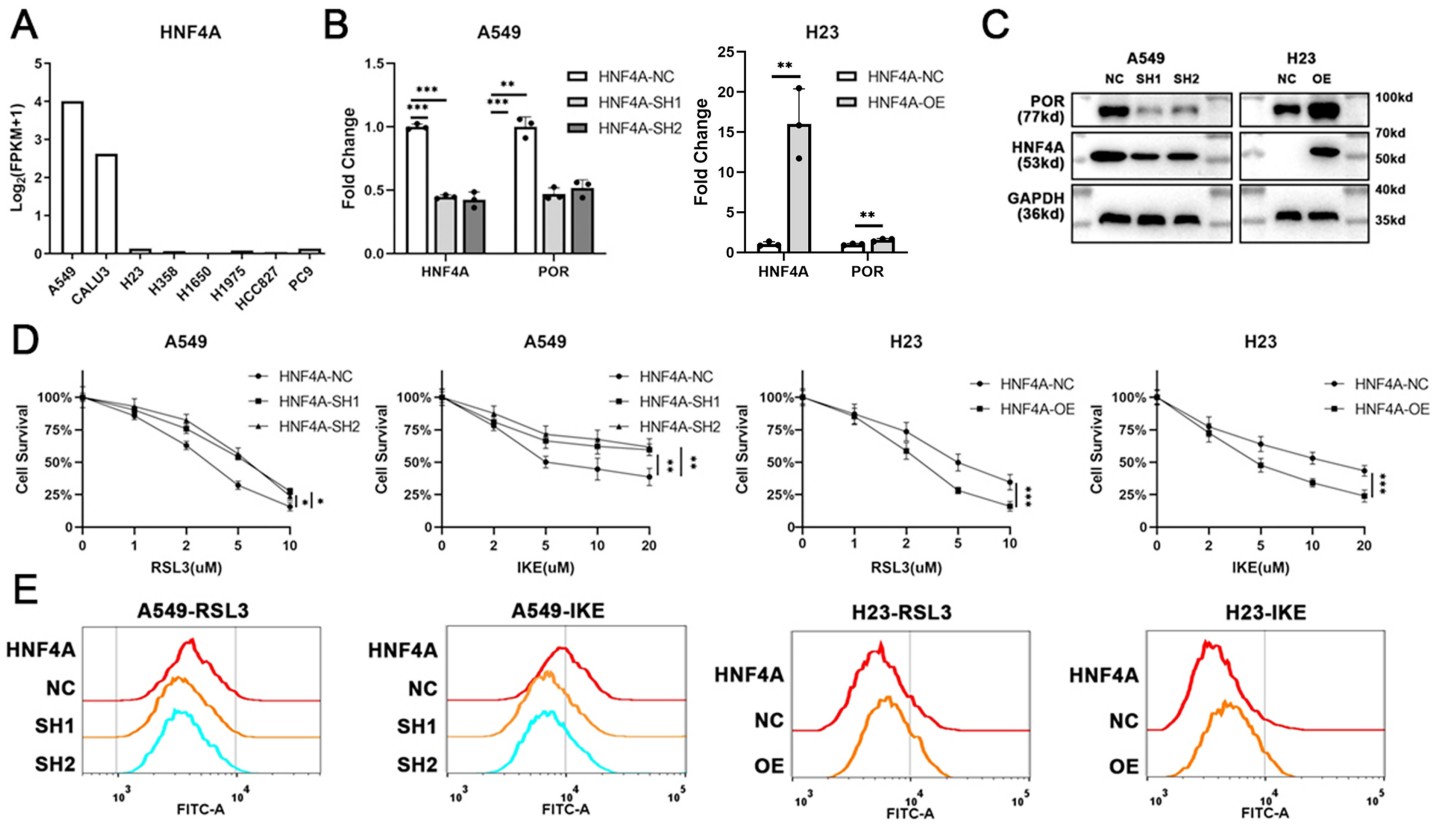

**Figure 2 HNF4A promoters ferroptosis and POR's expression in lung adenocarcinoma.** (A) The expression of HNF4A in different lung adenocarcinoma cell lines. (B) qRT-PCR and (C) Western blot results of HNF4A and POR expression in HNF4A knocked down A549 and HNF4A overexpressed H23 cells. (D) Cytotoxicity of RSL3 and IKE, and (E) the level of cellular lipid peroxidation in cells with knocked down or overexpressed HNF4A. $*p < 0.05$, $**p < 0.01$, $***p < 0.001$.

POR's promoter region and the other lacking different HNF4A binding sites (Fig. 3D). According to our findings, the expression of firefly luciferase with the POR's promoter was considerably suppressed by HNF4A knockdown and significantly enhanced by HNF4A overexpression (Fig. 3E). The promoting effect of HNF4A on POR expression was mostly restored when both two HNF4A binding sites were mutated (Fig. 3E).

## POR restoration blocked HNF4A's promotion on ferroptosis in lung adenocarcinoma

To demonstrate that HNF4A promoted ferroptosis *via* POR in lung adenocarcinoma, we overexpressed POR in HNF4A-knockdown A549 cells, and knocked down POR expression in HNF4A-overexpressed H23 cells (Figs. 4A and 4B). We found that POR overexpression considerably prevented the influence of HNF4A-knockdown on the cytotoxicity of ferroptosis inducers as well as the level of cellular lipid peroxidation in A549 cells (Figs. 4C and 4D). Similar results were also observed in experiments in H23 cells (Figs. 4C and 4D). These results validated that HNF4A promoted ferroptosis *via* POR in lung adenocarcinoma.

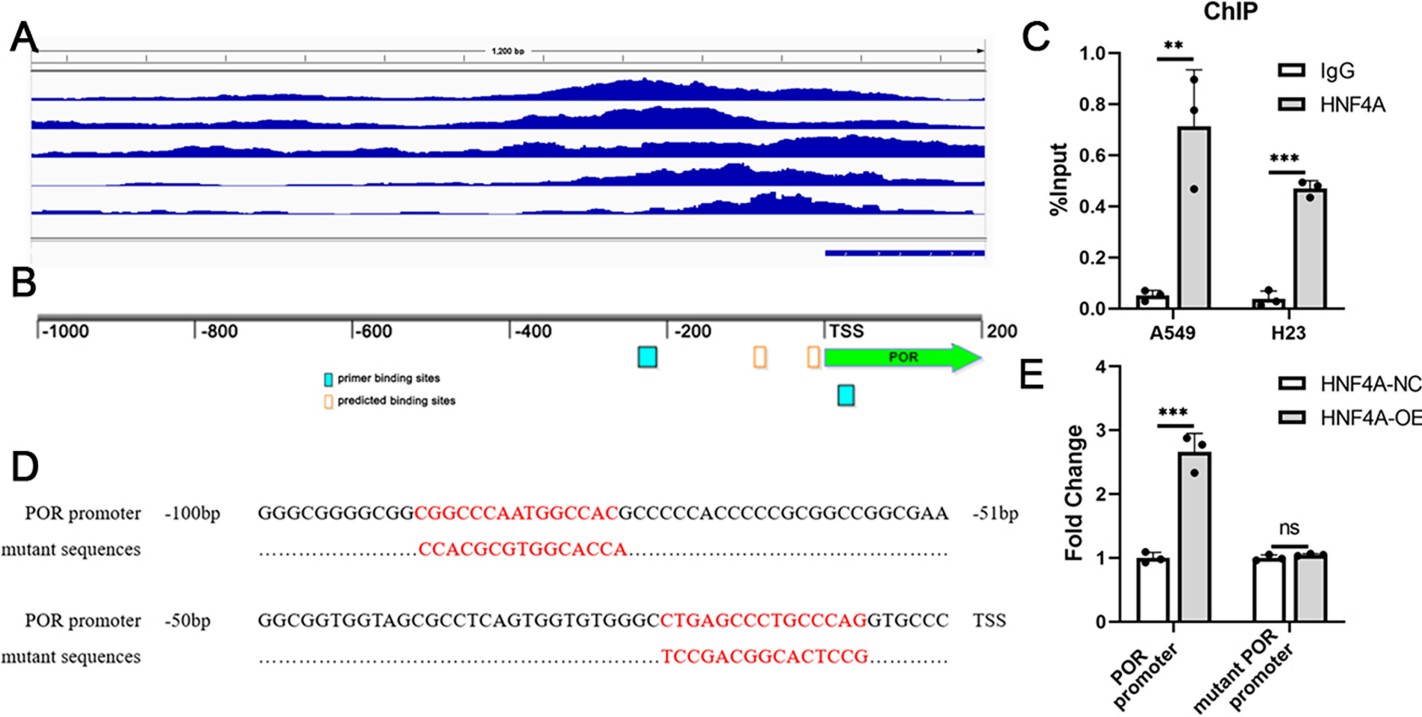

**Figure 3 HNF4A promoted POR expression *via* binding to POR's promoter.** (A) HNF4A binding peaks in POR's promoter according to the ChIP-seq data in the Encode database. (B) The locations of HNF4A's potential binding sites and primers in POR's promoter. (C) The ChIP-qPCR results. (D) the sequences of HNF4A's binding sites and corresponding mutated sequences in the luciferase assays. (E) The results of dual-luciferase assays. ns, not significant; **$p < 0.01$, ***$p < 0.001$.

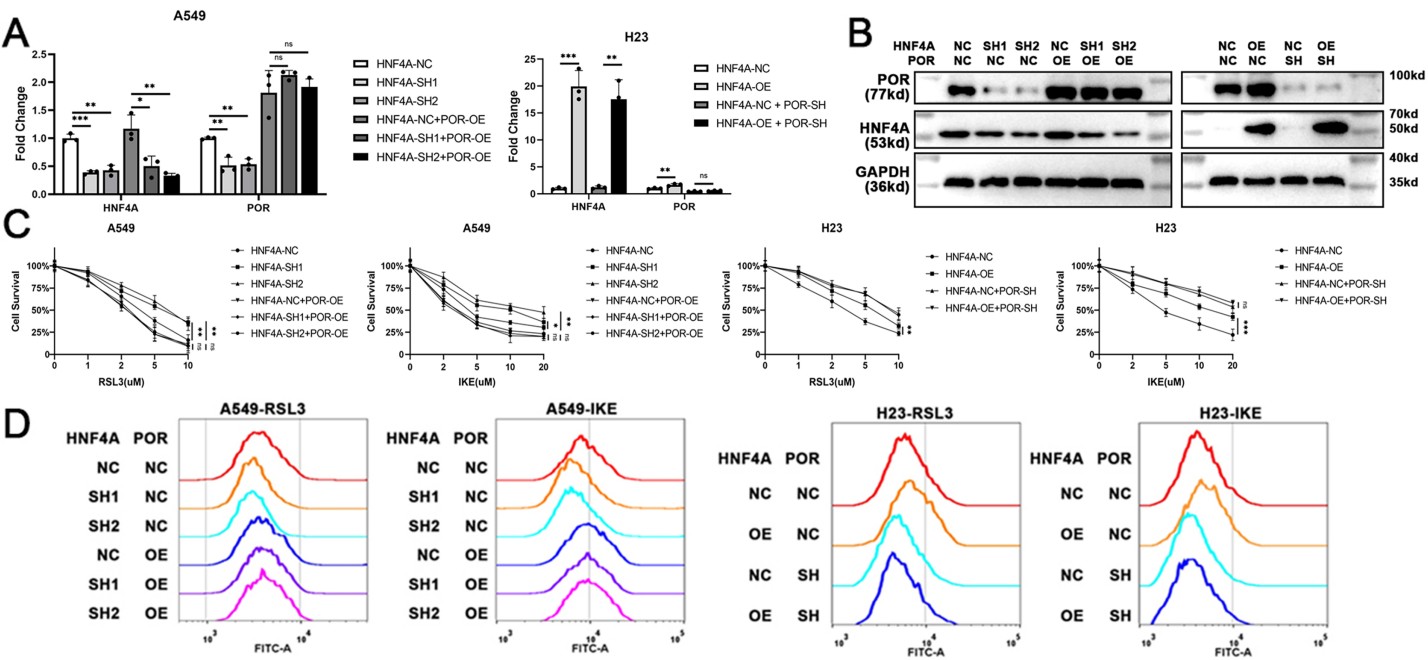

**Figure 4 POR restoration blocked HNF4A's promotion on ferroptosis in lung adenocarcinoma.** (A) qRT-PCR and (B) western blot results of POR restoration in cells with knocked down or overexpressed HNF4A. (C) The cytotoxicity of ferroptosis inducers and (D) the level of cellular lipid peroxidation in HNF4A-altered cells after POR was restored. ns, not significant; *$p < 0.05$, **$p < 0.01$, ***$p < 0.001$.

## DISCUSSION

It has not yet been shown how HNF4A affects POR expression or ferroptosis in lung adenocarcinoma. In this study, we found that ferroptosis inducers RSL3 and IKE significantly decreased the expression of HNF4A in A549 cells. We, therefore, hypothesized that HNF4A was involved in the regulation of ferroptosis sensitivity. This hypothesis was validated in cells overexpressing and knocking down HNF4A, and the results showed that HNF significantly promoted the ferroptosis of lung adenocarcinoma cells. Using the ChIP-seq data in the Encode database, we identified that HNF4A regulates POR, which is crucial for ROS production and ferroptosis, and validated it in cell experiments. The regulation of HNF4A to ferroptosis can be blocked by restoring POR. By using ChIP-qPCR and dual luciferase studies, we were able to better define the HNF4A binding site on the POR's promoter region.

There is some debate concerning HNF4A's role in malignancies. HNF4A was found to induce carcinogenesis and the development of cancers, according to several studies. For instance, *Chen et al. (2021)* found that the HNF4A aided in the growth and spread of invasive mucinous lung cancer *via* the HNF4A-BC200-FMR1 positive feedback loop. According to the findings of *Chang et al. (2016)*, HNF4A was a targetable oncoprotein in gastric cancer. They discovered that AMPK signaling controlled HNF4A, and HNF4A controlled the "metabolic switch" feature *via* targeting WNT5A (*Chang et al., 2016*). *Pan et al. (2020)* reported that HNF4A promoted gastrointestinal adenocarcinomas by activating the transcription of a number of downstream targets, such as HNF1A and several interleukin signaling proteins.

However, several other studies claimed that HNF4A functioned as a tumor suppressor in cancers. For example, *Yao et al. (2016)* found that HNF4A prevented EMT *via* the Wnt/β-catenin signaling pathway, and that HNF4 downregulation in colon cancer might be caused by the methylation in its promoter. According to *Gao et al. (2019)*, in renal cell carcinoma, HNF4A inhibited cell migration and invasion through transcriptionally controlling E-cadherin. *Ma et al. (2022)* reported that HNF4A reduced cervical cancer cell growth and tumor development by suppressing the Wnt/β-catenin pathway's activity. As a transcription factor, HNF4A may participate in the transcription of numerous target genes. Consequently, the varying significance of different target genes in various molecular backgrounds may confound HNF4A's function.

There is still much to learn about HNF4A's function in ferroptosis. According to *Zhang et al. (2019)*, the majority of ferroptosis-downregulated genes in liver cancer were regulated by HNF4A, and dissociation of KAT2B prevented HNF4A from binding to the promoters of these genes. *He, Li & Yu (2021)* discovered that HNF4A may function as a critical gene in ferroptosis using bioinformatics analysis. By promoting the expression of POR, a crucial gene for ROS production, our results demonstrated that HNF4A enhanced ferroptosis in lung adenocarcinoma, shedding light on the function of HNF4A in ferroptosis. According to *Luo et al. (2019)*, HNF4A was in charge of the hepatocytes' differentiation process's overall increased level of mitochondrial, peroxisomal, and non-mitochondrial oxidative metabolism. However, HNF4A, reported by *Darsigny et al. (2010)*, was able to eliminate

abnormal epithelial cell resistance to ROS generation during the development of intestinal tumors. Further research is still needed to determine how HNF4A affects ferroptosis in various tumor types.

POR, an endoplasmic reticulum membrane oxidoreductase, provides microsomal P450 enzymes with electrons straight from NADPH, which is necessary for numerous metabolic processes (*Zhao et al., 2021*). In a variety of cancer cell lines, POR deficiency greatly decreased the amounts of peroxidized polyunsaturated fatty acid-containing phospholipid under ferroptosis stress, and hence promoted robust resistance to ferroptosis caused by various stimuli (*Koppula, Zhuang & Gan, 2021*; *Yan et al., 2021a*; *Zou et al., 2020*). The majority of cancer cells express POR at high levels, suggesting its function as a general driver of lipid peroxidation (*Zou et al., 2020*). Our research indicates that a variety of variables, including HNF4A, may influence ferroptosis by controlling POR.

Our research has several limitations. First and most important is that we only performed cell experiments due to limited experimental conditions and funds. If clinical specimens or animal models were used our results would be more convincing. Second, there might be other targets of HNF4A involved in ferroptosis besides POR, but we failed to fully explore them. Detecting expression profiles of cells overexpressed or knockdown HNF4A using RNA-Seq will be much helpful. We hope these limitations can be addressed in the future.

## CONCLUSIONS

HNF4A promotes POR expression through binding to the POR's promoter, and subsequently promotes the ferroptosis of lung adenocarcinoma.

### Funding

This work was supported by the Natural Science Foundation of Shanghai: 22ZR1411900, and the Biomedical Technology Supporting Foundation of Shanghai: 22S11900300. The funders had no role in study design, data collection and analysis, decision to publish, or preparation of the manuscript.

### Grant Disclosures

The following grant information was disclosed by the authors:
Natural Science Foundation of Shanghai: 22ZR1411900.
Biomedical Technology Supporting Foundation of Shanghai: 22S11900300.

### Competing Interests

Jiaqi Liang & Cheng Zhan are Academic Editors for PeerJ.

### Author Contributions

- Valeria Besskaya performed the experiments, prepared figures and/or tables, and approved the final draft.

- Huan Zhang performed the experiments, analyzed the data, prepared figures and/or tables, and approved the final draft.
- Yunyi Bian performed the experiments, prepared figures and/or tables, authored or reviewed drafts of the article, and approved the final draft.
- Jiaqi Liang performed the experiments, analyzed the data, prepared figures and/or tables, authored or reviewed drafts of the article, and approved the final draft.
- Guoshu Bi analyzed the data, authored or reviewed drafts of the article, and approved the final draft.
- Guangyao Shan analyzed the data, authored or reviewed drafts of the article, and approved the final draft.
- Cheng Zhan conceived and designed the experiments, authored or reviewed drafts of the article, and approved the final draft.
- Zongwu Lin conceived and designed the experiments, authored or reviewed drafts of the article, and approved the final draft.

## Data Availability

The raw results of western blots, qRT-PCR, cell cytotoxicity, and luciferase assays are available in the Supplemental Files.

## Supplemental Information

Supplemental information for this article can be found online at http://dx.doi.org/10.7717/peerj.15377#supplemental-information.

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
