# Peer review of "Hepatic nuclear factor 4 alpha promotes the ferroptosis of lung adenocarcinoma via transcriptional activation of cytochrome P450 oxidoreductase"

_PeerJ, doi:10.7717/peerj.15377_

## Round 0.1 · original submission · Minor Revisions

The relationship between HNF4A and lung adenocarcinoma is worth studying. The manuscript should discuss the limitations of the study and include in vivo studies in the mechanism discussion or explain why only in vitro studies were conducted. In addition, mistakes in grammar and spelling should be avoided.

Reviewer 1 ·

Basic reporting

The authors found that HNF4A promoted POR expression through binding to the POR promoter, a key gene involved in ferroptosis, and therefore promoted the ferroptosis of lung adenocarcinoma. The role of HNF4A in lung adenocarcinoma is rarely reported, and its regulation effect on POR or ferroptosis has not been revealed before.

Experimental design

The authors overexpressed or knocked down HNF4A in several cell lines of lung adenocarcinoma, and then investigated the alteration of ferroptosis sensitivity and the expression of POR. The experimental design is OK.

Validity of the findings

The authors used restore experiments to validate the role of POR in HNF4A-regulated ferroptosis. But all experiments were performed in cell lines of lung adenocarcinoma. The authors should better validate their results in clinical specimens.

Additional comments

Although the overall English level of the article is well, there are still some grammatical errors, as follows.
In the abstract section:
In Line 26, "dual-luciferase assays was performed" should be "dual-luciferase assays were performed".
In Line 32, "POR serve" should be "POR served".
In Line 38, "the POR promoter " should be " the POR's promoter".
In the introduction section:
In Line 45, "which make up" should be "which makes up".
In the Materials and Methods section:
In Line 79, "Cells grown" should be "Cells were grown".
In Line 94, "as reviously described" should be " as previously described ".
In Line 116, " Cells was lysed" should be " Cells were lysed".
In Line 124, " To examined" should be "To examine".
In the results section:
In Line 155, "to detected" should be " to detect".
In Line 195-196, "there was several binding peaks" should be "there were several binding peaks".
In Line 199, "HNF4A bound to" should be "HNF4A was bound to ".
In the discussion section:
In Line 228, " It has not yet been showed" should be " It has not yet been shown".
In Line 254, " may participated" should be "may participate".
In Line 281, "the POR promoter " should be " the POR's promoter".

Reviewer 2 ·

Basic reporting

The authors mainly focused on the effects of a transcript factor called hepatocyte nuclear factor 4 alpha (HNF4A). They found that HNF4A dramatically increased cytochrome P450 oxidoreductase (POR), which is a key gene in ferroptosis, which in turn significantly accelerated ferroptosis in lung adenocarcinoma. There is some novelty in this manuscript. Raw data have been shared. Relevant literature has been cited and discussed.

Experimental design

The authors investigated whether the cells' sensitivity to ferroptosis inducers changed or not after HNF4A's expression was overexpressed or knocked down. Then the authors used ChIP-Seq data from the ENCODE and CTRP databases to lock in the downstream genes of HNF4A, and they found POR. Then the authors explored how HNF4A regulated the expression of POR using ChIP-Seq and dual-luciferase report assays, and restored POR to validate that HNF4A promotes ferroptosis via POR. This research is within the aims and scope of the journal. But how the authors analyze the data from the ENCODE and CTRP databases should better be more detailed.

Validity of the findings

Two cell lines were used in this study. The results were relevant to their scope. However, all the results were based on cell experiments. If the authors validate their results in vivo, the results will be more convincing. The authors should better discuss it.

Additional comments

(1) The authors should better clearly discuss the limitations of their research in their manuscript.
(2) Some grammatical errors should be revised. The authors should better carefully check their manuscript. For example, in the abstract, in "dual-luciferase assays was performed", "was" should be revised to "were"; in "POR serve as a potential target gene of HNF4A", "serve" should be past tense.

---

## Round 0.2 · accepted · Accept

The authors have addressed the concerns of the reviewers.